# Innovative DMHS Algorithm Application in Wireless Sensor Networks for Efficient Routing in High-Risk Environments

**DOI:** 10.3390/s23167223

**Published:** 2023-08-17

**Authors:** Yuanjia Ma, Xiangwu Deng

**Affiliations:** School of Electronic Information Engineering, Guangdong University of Petrochemical Technology, Maoming 525011, China; dengxiangwu2019@gdupt.edu.cn

**Keywords:** routing, unreliable links, dynamic programing, WSNS, black hole

## Abstract

Efficient routing is essential for the proper functioning of wireless sensor networks (WSNs). Recent research has focused on optimizing energy and delay for these networks. Nevertheless, there is a dearth of studies that have examined the effects of volatile settings, such as chemical plants, coal mines, nuclear power plants, and battlefields, where connectivity is inconsistent. In such contexts, sensor networks may face security incidents, and environmental factors such as node movement and death can result in dynamic changes to the network topology. A novel design algorithm grounded on Dynamic Minimum Hop Selection (DMHS) was introduced in this paper. The key principle behind DMHS is to use a probabilistic forwarding decision-making process through a distributed route discovery strategy that utilizes dynamically adjusted minimum hop counts of nodes. Simulation results indicate that the life cycle of the DMHS algorithm increases by more than 12% over 700 nodes when compared to the traditional energy-saving algorithm. Furthermore, our algorithm performs better in the average delivery rate of node, and has a 10% to 21% improvement compared to the other algorithms. Overall, the DMHS algorithm represents an important contribution to the development of WSNs that can function robustly in high-risk and unstable environments.

## 1. Introduction

WSNs are a cost-effective and versatile solution for military, environmental, disaster relief, and industrial safety applications. However, WSNs face significant challenges in routing and energy consumption. While considerable research has addressed these issues in the past decade, insufficient attention has been given to their robustness. Given the intricate nature of the environments in which WSNs operate, they require not only energy efficiency but also high levels of stability and resilience. To fully appreciate these challenges, two aspects deserve attention.

In high-risk circumstances, WSNs are more prone to encounter “accidental holes”. These are referred to as traditional WSN holes, where key nodes in sensor networks may become exhausted due to overuse, resulting in the network’s “energy hole” or “black hole”. Research has been focused on addressing the energy hole issue in WSNs by employing power control [1,2,3] and topology control techniques [4,5,6]. This involves utilizing control mechanisms based on perception models and implementing heterogeneous distribution strategies to delay node failure and extend the overall network lifespan.

Additionally, some research has proposed black hole avoidance mechanisms in routing algorithms, such as energy balancing algorithms [7,8,9], routing algorithms based on location information [10], and planarization routing algorithms [11]. These algorithms can partially solve the energy hole problem in WSNs. However, the potential for “event holes” has yet to be fully considered. In other words, there may be emergency cases occurring such as explosions in industrial environments, even if the odds of a calamity are low. Hence, it is of paramount importance to formulate efficient strategies that encompass not solely the issue of energy holes in WSNs, but also encompass the possibility of unanticipated circumstances that could potentially disrupt the network’s operational efficacy. By utilizing a combination of strategies, the life of the WSN can be prolonged, thereby ensuring that data transmission remains stable and reliable in critical circumstances.

In the event of a catastrophic incident, it is common for the network topology to experience frequent changes. For instance, hazardous situations such as explosions, fires, or radiation leaks can inflict extensive damage upon predefined nodes. To facilitate accident response efforts, robots or unmanned aerial vehicles are deployed as temporary mobile nodes at the incident site to acquire local data. The incessant modifications to the network topology result in a proliferation of routing discovery packets, leading to data conflicts and network congestion. This monitoring system typically operates on a low duty cycle, with minimal data transmission during normal circumstances. However, once an accident occurs, a significant volume of data needs to be transmitted, rendering the routing problem more intricate due to the energy constraints imposed on the nodes, which distinguishes it from a typical WSN system.

Therefore, it is imperative to devise an algorithm capable of deployment in hazardous settings such as chemical plants, coal mines, nuclear power plants, and battlefields. In such circumstances, sensor nodes are highly prone to failure, necessitating the development of a novel routing algorithm to guarantee robustness.

There are scholars who have conducted research on routing algorithms for WSNs in the context of unreliable links. These studies share similarities and differences with the application scenarios of the current study. The similarity is that the scenarios involving unreliable links are all a result of topology changes caused by energy exhaustion or node mobility, known as Mobile ad-hoc networks (MANETs). For instance, El-Fouly, Fatma H et al. [12] expound on the successful application of dynamic programming to address the issue of unreliable links, which provides strong empirical support for our theory. In addition, Poluboyina L et al. [13] evaluate the performance of various unicast routing protocols through a simulated comparison in a high-risk environment. However, this paper focuses on the routing strategy for safety monitoring WSNs operating in hazardous environments where links are unreliable with sudden death at any time. Drawing upon the principles of dynamic programming, a distributed routing discovery strategy is employed. Additionally, a self-adaptive flooding optimization algorithm is proposed for WSNs, addressing the issue of network load imbalance.

The main contributions of this paper are summarized as follows:This paper presents a robust routing algorithm for extreme environments, such as petrochemical plants, field scout facilities, and disaster relief operations. The algorithm selects the minimum number of hops based on dynamic programming principles. It utilizes a distributed routing discovery strategy that incorporates probabilistic forwarding decisions and dynamically adjusts the minimum hop selection of nodes. Simulation results demonstrate that this algorithm enhances network stability and reliability in high-risk environments, even when a large number of nodes become invalid, resulting in a massive black hole in the network.In contrast to conventional routing algorithms, the method presented in this paper can be applied to sensor networks that lack a location system and have a three-dimensional structure. Examples of such networks include atmospheric monitoring in space and underwater WSNs. Furthermore, experimental results from a designed instance demonstrate that the method exhibits enhanced flexibility and robustness in the presence of unreliable links.

In Table 1, we make a comparison of the more commonly used algorithms, pointing out their advantages and disadvantages.

## 2. Related Works

The routing problem in WSNs entails determining the most efficient route from the monitoring region to the sink node. This route is essential for timely detection and monitoring of regional data, as well as for conducting associated processing. The optimal path takes into consideration not only the shortest Euclidean distance but also factors such as minimal energy consumption or minimal transmission delay for data. Each edge in the network is assigned a non-negative constant weight that represents its importance. Consequently, finding the optimal path is tantamount to identifying the shortest path.

Currently, there are several types of shortest path algorithms available, including the Dijkstra algorithm [14], matrix algorithm [15], greedy algorithm [16], and Floyd algorithm [17]. The Dijkstra algorithm is a well-known single-source shortest path algorithm. While it can yield the optimal solution for the shortest path, its efficiency is relatively low due to the extensive calculations required for node traversal, resulting in a time complexity of 2O(n). On the other hand, the matrix algorithm can solve the shortest path between all pairs of nodes, but it necessitates significant computation, making it more suitable for computer-based calculations. Its typical time complexity is 3O(n). The greedy algorithm, although simple and feasible, only provides a local optimal solution in terms of routing and decision-making. As for the Floyd algorithm, it is primarily utilized in directed graphs and can compute the shortest distance between any two nodes. Nonetheless, its high time complexity of 3O(n) makes it unsuitable for large-scale data calculations.

As WSNs are increasingly being applied in various real-world scenarios, the reliability of these networks has become a key concern. Therefore, the robustness of a routing algorithm is an important performance metric to consider, alongside factors such as energy consumption and communication delay. In high-risk environments such as petrochemical plants, a leakage incident can potentially lead to explosions. Sensor nodes are constantly at risk of failure, which can result in the formation of a “black hole”. Although algorithms such as Two-Phase Geographic Greedy Forwarding (TPGF) can partially bypass these holes, they are often prone to becoming stuck in local minima, as illustrated in Figure 1. Excessive node failure necessitates each node to repeatedly search for new paths, resulting in energy wastage.

To solve the routing problem in harsh environments, a considerable amount of research has been carried out during the last decade. Kavra R et al. [18] propose a planarization routing algorithm based on GG and RNG at the worst case. The findings indicate that within certain planar topology control schemes, it is feasible to recover from a failure in greedy routing without needing to switch between any adjacent faces. Guo S et al. [19] introduced an opportunistic flooding approach specifically designed for low-duty-cycle networks characterized by unreliable wireless links and predetermined working schedules. The main concept involves making probabilistic forwarding decisions at the sender based on the delay distribution of next-hop nodes. In recent years, considerable research efforts have been devoted to tackling the routing problem in demanding environments. Among the notable contributions, a noteworthy study conducted by Sateesh Gudla et al. [20] put forth a genetic algorithm-based approach for energy-efficient data collection. This algorithm effectively mitigated the issues of excessive data transmissions, high energy consumption, and network packet delivery delays, while also extending the network’s lifetime. Moreover, the study demonstrates the efficacy of the proposed method through the evaluation of essential parameters such as energy consumption, network lifetime, number of data transmissions, and average delivery delay, thus providing comprehensive evidence of its effectiveness.

In the context of low-duty-cycle networks with unreliable wireless links and predetermined working schedules, Cheng L et al. [21] proposed a new energy-efficient adaptive forwarding technique called EEAF. It demonstrated the presence of path diversity in low-duty-cycle WSNs, where routing for optimal delay and optimal energy consumption are likely to follow distinct paths. In study [22], Tang J et al. introduce FlowerCast, a multicast protocol that reduces multicast delay and improves the delivery ratio by quantifying communication distance, constructing a delay-optimal multicast tree, and employing a hybrid routing strategy to maximize the potential of overlapping links.

Sabri Y et al. [23] proposed a theoretical model that calculates the probability density function of multi-hop broadcast latency in WSNs when employing probabilistic broadcasting schemes. The study introduces a new probabilistic approach for directed data transmission that eliminates the need for route discovery. The proposed model ensures that each message reaches the base station (BS) successfully with a specified success probability.

In another study by Suma S et al. [24], the authors investigated the protocol under consideration utilizes an energy-conscious routing algorithm to identify the most efficient routing tree connecting the Cluster Heads (CHs) to the sink. This algorithm designates specific paths for data transmission towards the sink while periodically adjusting them to ensure a balanced energy consumption among nodes and enhance the overall network longevity.

## 3. Network Model and Problem Statement

In this section, we define the network model and assumptions related to our dynamic minimum hop selection as follows.

### 3.1. Network Model

The objective of routing in WSNs is to pinpoint the most effective route from the source node to the sink node. This challenging task is simplified by the application of dynamic programming, which breaks down the compound multi-phase problem involving N processes into individual single-phase problems. It bases its assumption on the logic that optimizing every step independently culminates in the selection of the best N process as the outstanding choice. Hence, dynamic programming proves to be an effective strategy to address routing challenges in WSNs. A practical example is visualized in Figure 2, which illustrates a dynamic programming routing model composed of an n + 1 level, drawing reference from the study [25].

Consider a WSN consisting of N sensor nodes, uniformly distributed across a square plane region. This network is defined by the following characteristics: it contains a single sink node, represented as v0. The sink node and the source node can both be arbitrarily located within the WSNs. Moreover, each sensor node can fall under one of three states: active, available, or non-functional. The WSNs under consideration can be represented as a graph G(v, C), where v = {v_1_, …, v_h+1_} represents a finite set of sensor nodes with hop count number of h, and nodes are identified by V_0_, V_1_, …, V_h+1_ in every level. In order to distinguish between each node in place at all levels, we used V_hi_ to express node i with the minimal hop h. The V sets divide the path into h + 1 stages represent as P. The set C = {C_(11,0)_, …, C_(h+1,hi)_} is a finite set of edged links weight, which is determined by the hops and remaining energy of nodes.

Di is defined as the set of adjacent nodes to a given node i. If node j ∈ D_i_ and i ∈ V_h_, then it follows that D_i_ ∈ V_k−1_. The routing decision at node i is based on the metric between i and the nodes in D_i_. During routing, only links that meet the minimum hops requirement are evaluated, and the probability of selecting a node depends on its remaining energy. This mechanism balances energy consumption and link robustness to ensure optimal network performance.

### 3.2. Energy Modeling for Wireless Communications

The wireless communication of sensor nodes involves various aspects of energy consumption, such as signal amplification energy, signal transceiver energy, and data processing energy. In this paper, we utilize the more classical energy consumption model [26] to evaluate the energy consumption of nodes.

In WSNs, the data transmission phase constitutes a significant energy consumption component. The energy consumed for sending the same data packet theoretically varies based on packet size. To minimize energy consumption during data transmission and ensure the minimum energy is utilized for packet sending, this study use the optimal packet size, denoted as P_opt_. The concept of P_opt_ has been previously discussed in the literature [27], and its calculation is presented in Equation (1):(1)$$P_{\mathrm{opt}}=\frac{\sqrt{C_{0}^{2}-\frac{4C_{0}}{\ln(1-p)}}-C_{0}}{2}$$
where C_0_ = α + K_2_/K_1_, α denotes the packet header size (bit), K_1_ denotes the energy consumed by the payload in communication, K_2_ denotes the energy consumed by the node for the startup, and p denotes the bit error rate (BER) of the channel. Thus, the energy consumed by the node to send and receive P_opt_ bit data E_TX_ and E_RX_ are shown in Equations (2) and (3), respectively:(2)$$E_{\mathrm{TX}}(P_{opt},x)=P_{opt}\times E_{\mathrm{elec}}+P_{opt}\times \varepsilon_{\mathrm{amp}}$$
(3)$$E_{\mathrm{RX}}(P_{opt})=P_{opt}\times E_{\mathrm{elec}}$$
where x denotes the distance between sensor nodes, E_elec_ denotes the energy consumed to send or receive a unit bit, and ε_amp_ denotes the energy consumed to amplify the signal of the transmitting node. Thus, the P_opt_ bit sized packet is sent from send to receive and the total energy consumed E_total_ is shown in Equation (4):(4)$$E_{\mathrm{total}}=P_{\mathrm{opt}}\times (2E_{\mathrm{elec}}+\varepsilon_{\mathrm{amp}})+E_{\mathrm{DA}}$$
where E_DA_ denotes the energy consumed by the data during fusion at the cluster head node. For simplicity, we do not consider the energy consumed by data fusion, so this E_DA_ we set to 0 in the simulation.

### 3.3. Assumptions

In this section, we will outline a set of fundamental assumptions that a network scheme must satisfy:The sink node is located at a fixed position within region A and has abundant energy. It can retrieve the ID and hop count of each node. Additionally, each node can ascertain its minimum hop distance from the sink node.All nodes have the same communication radius, and the links between them are symmetrical.Nodes may experience sudden death or massive failure due to energy depletion or field accidents.Rescue nodes carried by drones or robots can join the WSNs at any time. These nodes are referred to as mobile nodes.The communication radius of nodes is limited, and the convergence nodes can only be reached through multi-hop networks. Only a few nodes can move and will not move frequently.

### 3.4. Format of Data Packet

The routing protocol proposed in this study selects routes based on H value and task-type, which necessitates modifications to the format of data packet. Table 2 presents the modified routing table.

The packet consisted of a head and body. The header of a packet consists of five parts, as shown in Table 2. The ID represents the node number where the packet has already arrived. The IDs and IDd represent the ID numbers of the nodes from which the packet is forwarded and to which it is forwarded, respectively. The H field represents the current hop count from the node to the sink, which is initialized to the minimum hop count. The flag bit is used to trace back the previous node when all the next hop nodes fail. The field of tasktype gives the types of tasks, and it consists of four kinds of normal, emergency, join, and quite task type. Src and Dest refer to the source and destination, respectively, indicating the initial source node ID and the ultimate destination node ID of the packet. Due to the critical role of precise hop counting in wireless sensor networks for determining the distance or proximity between nodes, any corruption or miscalculation of the hop count can have detrimental effects on the algorithm’s performance.

The purpose of including the checksum field in the packet is to enable error detection during transmission. The checksum field consists of two bytes, one of which uses parity to reduce the likelihood of errors. The other byte is used to store the original H (hop count) value of the node. If any corruption or miscalculation of the hop count occurs during transmission, the H-value of the node is restored to its original value. Additionally, the node broadcasts a frame to all its neighboring nodes, instructing them to update their routing table. This ensures that the node can rejoin the network and resume normal communication.

## 4. Methods and Design

The primary objective of this paper is to develop a routing algorithm for WSNs that possesses robustness, low latency, and energy efficiency. This algorithm should be suitable for deployment in high-risk environments. Specifically, our aim is to design a method that can be utilized by a supervisory system during emergency situations, while also ensuring a low packet loss rate and energy efficiency.

The decision-making process in routing includes determining the next hop and comprises three phases: establishing link weights, applying a dynamic selection algorithm for minimum routing hops in emergency scenarios, and implementing inclusion and removal mechanisms for nodes. The problem has been transformed into a set of static subproblems by utilizing minimum hop paths determined through a status map. Previous research has indicated that choosing routes with the least number of hops does not necessarily result in minimal end-to-end delay. Thus, it is crucial to establish a standard for selecting subsequent hops in each stage. The DMHS routing model takes both robustness and low energy consumption into consideration in order to achieve optimal transmission and maintain a balance between routes with minimal costs.

### 4.1. The Initialization Link of the Weights

The initial step in this paper involves the configuration of initial weights for link establishment. These weights are determined by the number of hops and remaining energy available. The first phase involves establishing the minimum hop. The sink node initiates a flooding packet to all neighboring nodes containing the hop count (H) between itself and the sink node. Upon receiving the flooding packet, each neighbor compares its current hop count (initialized to the maximum value) with the received H + 1 and assigns the smaller value to its hop count. Nodes with the same hop count and next hop are stored in an array, and the others are discarded to prevent a broadcasting storm. This paper provides further details on this process in Figure 3.

The probability mass function is a quantitative measure used to determine the probability of node selection for the purpose of energy balancing. In this paper, the probability is not solely defined based on the lowest cost but also considers the tradeoff between energy consumption and link robustness in the routing process. By considering the selections made at each hop, we can establish a routing path from the source node *V_h_*_+1_ to the destination node *V*_0_. The objective of this paper is to identify a route that not only satisfies the energy and latency requirements but also possesses strong robustness. In other words, our aim is to find a route that optimizes the overall sum of *C_k_*_(*j,i*)_, represented as the optimal cost *C_opt_*_(*j,i*)_.
(5)$$\left\{\begin{array}{c} C_{opt}(V_{h+1},V_{0})=\min\{C_{k}(V_{h},V_{0})+C_{h+1}(v_{h+1},v_{h})\},1\leq k\leq h\\ C_{1}(V_{0},V_{0})=0 \end{array}\right.$$
where *C_k_*(*V_h_,V*_0_) represents the sub-routes from the sink node to any intermediate node, which are determined in each node. This helps to avoid unnecessary computations and the wastage of energy. *C_h_*_+1_(*v_h_*_+1_,*v_h_*) is a link between *V_h_*_+1_ and *V_h_*, and if its cost is optimal, the link will be removed. The selection of the suitable node *v*_*hi* is a probability function based on the remaining energy *E_hi_* and *E_h_*_+1_*i*, which we reformulate as follows:(6)$$p_{Hi}=\frac{E_{hi}(h+1-H)+\alpha E_{(h+1)i}(H-h)}{\sum E_{h}+\beta \sum E_{\left(h+1\right)}}\;i=1,2,\ldots n$$
where *P_Hi_* is the selection probability of node *Vi* with the minimum hop of *H*, and the candidates are elected from *V_h_* and *V_h_*_+1_, i.e., the value of *H* can be set to *h* + 1 or *h*, but nodes are more prone to selecting node from *V_h_*. *E_h_* and *E_h_*_+1_ is the remaining energy of nodes in set *V_h_* and *V_h_*_+1_, and it represents energy of *i*th node with the subscript of *i*. The parameter and, with the range from 0 to 1, are the weight coefficients of whether choosing the child nodes or neighbors with same hops. The larger the value, the stronger can local minima be avoided, but there must be a tradeoff between robustness and energy efficiency.

### 4.2. The Dynamic Minimal Hop Selection Algorithm

In this section, we will discuss strategies for ensuring network robustness during emergency situations. During emergencies, sensor nodes are vulnerable to sudden failures, which can lead to the creation of black holes. As these black holes expand, the availability of alternate routing paths diminishes. This expansion of black holes can ultimately result in the formation of isolated islands and network disconnection.

While flooding is known for its robustness in sensor networks, it can cause network congestion when multiple packets, including data and control signals, are transmitted simultaneously. This congestion can result in delays in data transmission and ultimately lead to network crashes.

To address these issues, this paper proposes a novel adaptive algorithm based on dynamic programming that searches for optimal paths in real-time. The concept of optimality in this context takes into account not only the shortest path, but also factors such as energy balance and time delay. The specific steps of the dynamic minimum hop selection algorithm for emergencies are as follows:

Step 1: Upon receiving emergency packets, the node responds with an ACK frame and checks if the ID matches its own. If it does, the routing is completed. Otherwise, proceed to step 2.

Step 2: Use heuristic functions to select a suitable next h − 1 hop node and wait for the ACK signal after forwarding the packet. If the ACK is received in a timely manner, the process can be completed. If not, proceed to step 3 in case of a timeout.

Step 3: The node experiencing a timeout checks if any other h − 1 nodes exist. If they do, return to step 2. If not, proceed to step 4.

Step 4: Increment the node to h++ and send a broadcast message to its neighbors for hop updating. Then, check if there are any h − 1 nodes. If there are, go to step 2. If not, return to step 4.

By transforming the multi-step decision process into a distributed routing discovery strategy through recursive calls from step 2 to step 4, the algorithm avoids infinite loops in step 4. This is because after multiple increments of h++, there will always be at least one node (its original h + 1 parent node) to return back to. Algorithm 1 presents the pseudo code of the dynamic minimum hop selection algorithm.
**Algorithm 1**: The pseudo code of the dynamic minimal hop selection algorithm.Input: Type of task, IDs (ID number of previous node), Dest (routing destination), h (hops to the sink of this node), ID (node Identifier), Flag (the initial value is 0).1:Return an ACK frame when an emergency task is received2:If $Dest\notin V_{h-1}$ then3:  Rec: if $V_{h-1}\neq \varphi$ then4:    $\mathrm{IDd}\leftarrow {\mathrm{select}\;a\;\mathrm{node}\;\mathrm{from}\;\mathrm{ID}}_{h}{\;\mathrm{and}\;\mathrm{ID}}_{\left(h-1\right)},\;\mathrm{IDs}\;\leftarrow \mathrm{ID}$
5:    forward frame, wait for the ACK6:      if timeout then7:        delete IDd from the of V_h−1_, goto Rec8:      else9:      return IDd10:      end if11:    else if Flag ≠ 212:     Flag++, h++, broadcast the hop change frame13:     goto Rec14:    else15:      Flag ← 0, drop the frame16:/* After two increments of h, there will always be at least one parent node to return to. If there is no parent node present, it indicates the existence of a disconnected node or a “lonely island”. In order to conserve energy, the frame is dropped. */17:end if18:else19:IDd←Dest, return IDd20:end if

### 4.3. Joining and Exiting Mechanism

The joining and exiting mechanisms described in this paper pertain to the dynamic updating of the topology of WSNs. When a new node, carried by either a robot or an unmanned aerial vehicle (UAV), intends to join the network, the node broadcasts a join frame to its neighboring nodes. The neighbors then respond by including their respective identifiers (h_1_, h_2_, … h_n_) in the ACK_Join frame sent back to the node.

Afterwards, the identifier (h) of the new node is set at min(h_n_)+1. Subsequently, the new node notifies its neighbors to update their routing tables and include the new node.

Alternatively, in situations where the energy of a node is insufficient, it has the capability to autonomously increment its identifier h to h + 1 and broadcast its hop count to neighboring nodes. Subsequently, other nodes will automatically evaluate the hop count of these nodes to determine if they should be avoided when making routing decisions.

In the event of a missing next hop node, a guaranteed return path can be established within a worst-case scenario following two hop count iterations. The node’s “h” value can be adjusted based on various factors, such as residual energy, link quality, energy balancing, channel quality, node faults, and mobility, to achieve optimal network performance.

### 4.4. Memory Requirements for the Routing Tables

The DMHS algorithm is a reactive routing protocol that establishes routes exclusively when packets necessitate transmission. This characteristic results in relatively modest memory requirements and low space complexity. However, the DMHS algorithm places some strain on time complexity as it requires the maintenance of routing table entries for active routes.In the DMHS algorithm, each routing table entry solely includes the next-hop data and the hop count towards the intended destination. Consequently, in a network with s destinations, each node might need to store n routing table entries. This implies that as the number of destinations in the network increases, the memory demand on each node also rises. For the purposes of this paper, we will focus solely on the scenario involving a single destination node. Moreover, because the DMHS algorithm is dynamic, the routing table must be updated whenever there are changes in the network topology, further escalating the memory requirements. Nonetheless, the DMHS algorithm generally demands less memory than link-state or distance-vector based routing algorithms since it excludes the need to store the topology information of the entire network. This grants the DMHS algorithm an advantage in resource-constrained wireless sensor networks.

### 4.5. Time Complexity Analysis

The implementation of the local flooding method provides the basis for establishing the minimum hop gradient table. Each node in the network not only identifies its immediate neighboring nodes as the next hop, but also stores information about adjacent nodes at the same hop level and the preceding node. This approach significantly minimizes the occurrence of latency and energy consumption caused by the frequent and transient querying of neighboring nodes in response to changes in network topology. Furthermore, when disseminating information, it is recommended to incorporate a random retreat period to mitigate collisions and energy expenditure.

In the initialization phase of the DMHS algorithm, each node needs to be checked and updated, so its time complexity is O(n*m) in the worst case for a graph containing n nodes and an average of m neighboring nodes. This is because in the route initialization phase, we need to find the node among the unvisited nodes that is 1 less than the current hop count and update the temporary distance of its neighboring nodes. Thus, for n nodes, the complexity of the algorithm is O(n). In addition, each neighbor needs to be considered once when notified to update the H-value, so the number of neighboring nodes also has an impact on the time complexity of the algorithm, which is O(m). Thus, the total time complexity is O(n*m).

Such complexity means the DMHS algorithm may face efficiency problems when dealing with large-scale graphs, especially when the density of the graph increases or when the communication radius increases, the number of neighbors may show a rapid growth. In this case, considering both energy consumption and link quality, the node employs a restricted flooding strategy to randomly select the h − 1 hop node as its A route can be established in the network by flooding a limited number of packets, while simultaneously addressing the broadcast storm problem and mitigating rapid failure through focused path exploration. This also reminds us that when designing and analyzing algorithms, in addition to their problem-solving ability, we should also consider their complexity to ensure their feasibility in real-world scenarios.

## 5. Experiment Setup

A series of actual field deployment experiments were conducted to verify the feasibility of the method proposed in this thesis. Due to limitations in the number of available sensors and the conditions of the experimental field, a total of 26 wireless sensors were utilized, with one serving as a sink node and another as a mobile node. For the beacon nodes, DMHS and other reference protocols were implemented using the TinyOS platform. Figure 4 displays a portion of the experimental setup. We arranged 24 of the beacon nodes outdoors, where the nodes were not exactly in the same plane in order to simulate the real environment as much as possible, and the batteries of the nodes were randomized and not purposely filled for them. The sink node was positioned at the network’s edge to mimic a large multi-hop sensor network, enabling testing with fewer sensors. It is important to note that all nodes were operated without antennas during the experiments, and the communication radius was approximately 10 to 15 m. If antennas are added, the sensor network can be deployed over a much wider range.

In order to visualize the routing process, we drew a schematic diagram based on the actual results, as shown in Figure 5. In this network, the sink node is represented by a solid black circle, and each node’s minimum hop distance to the sink is indicated by the number within its circle. Suppose a node is transmitting data with a hop count of 5, following the gradient of minimum hop counts shown in Figure 5a.

We simulate a node failure by turning off nodes with 1 hop, represented by the disappearance of circles, as shown in Figure 5b, the node with a hop count of 2 may be unable to locate its next node with a hop count of 1. In this case, the node will solve the subproblem recursively by increasing its hop count to 3 and continuing to search for nodes with a hop count of h − 1. As shown in Figure 5c, the 3-hop node successfully discovers the node with a hop count of 2 among its neighbors. If the node cannot find such a node (for instance, if the 2-hop node is also non-functional), it will further increase its hop count to 4 and continue the search until it finds a 3-hop node, as depicted in Figure 5d. By repeating this recursive process, the parent node with a hop count of 3 can ultimately solve the subproblem and find the routing path at the network’s top after a limited number of iterations. This algorithm is robust and effectively prevents the network from falling into a local minimum.

In an extreme case where the crucial 2-hop node is disabled, Figure 5e illustrates that the top portion of the network becomes disconnected. To bypass the black hole, the routing path forms a loop, with the hop counts of the nodes being updated along the way. Eventually, the path returns to the 3-hop node (actually a 5-hop node after two rounds of updating) at the crossroads, allowing the route to the sink to continue despite the obstacle. The annotation “+1” beside a node represents the number of updates, indicating how frequently the node’s hop count was adjusted.

Through dynamic adjustments, the hop count of nodes that are unable to pass or face difficulties in passing is increased. When the next packet arrives, it automatically selects the optimal path to avoid obstacles and unnecessary routing. Figure 5f demonstrates that when the next packet arrives, the source node directly chooses the more optimal path at that moment. The packet continually weighs the two paths during the dynamic adjustment process. Eventually, it becomes clear that the hop count of the source node needs to reach 8 for the path below to be ultimately determined. If some of the nodes in this path become overloaded or experience excessive energy consumption, they will adjust their hop count to achieve a balance in the optimal path.

Now let us consider the case where there are mobile nodes patching the network. In extremely harsh environments, such as a bombing disaster, it is possible that all the paths in the network fail, and it is necessary to use an unmanned aerial vehicle or a robot to transport new nodes to patch the network. We place mobile nodes at the edges of the network as shown in Figure 5g, where the mobile nodes are represented by blue pentagons. Through the joining and exiting mechanism described above, the mobile node will publish broadcast frames to notify the surrounding neighboring nodes to dynamically update the hop count. After a similar process as above, a new green path is finally established as shown in Figure 5h. Of course, this path may eventually run out of energy due to the large amount of data forwarding, but it is worthwhile for high-risk environments to return large amounts of live data in time.

## 6. Simulations and Performance Evaluation

Due to the limitations in the number of sensors and the site area, our practical deployment experiments only serve to verify the feasibility of the algorithms, and cannot provide an evaluation of their actual performance under conditions with a large number of nodes. Therefore, in the subsequent section, we will utilize simulation methods to assess and compare the performance of various algorithms. The following method will be employed:The impact of the number of nodes on network performance.Compare the network lifetimes of DMHS, MHR, Dijkstra, and TPGF.Investigate the effect of the percentage of failed nodes on network connectivity.The impact of death rate to the average delivery rate.

### 6.1. Experiment Setup

In order to validate the aforementioned algorithm, we constructed a WSN scenario measuring 1000 m × 1000 m using NS2. We compared the performance of our algorithm against the TPGF algorithm and the opportunistic flooding algorithm by varying the number of nodes. The distribution of nodes in the simulation environment was randomized. The sink had an ample power supply, and each node was initialized with a constant energy value. As stated in [28], the energy consumption of nodes is dependent on the data length (*k*) and distance (*d*).
(7)$$E_{s}(k,d)=\left\{\begin{array}{c} kE_{elec}+k\varepsilon_{f}{}_{s}d^{2}d<d_{0}\\ kE_{elec}+k\varepsilon_{mp}d^{4}d>d_{0} \end{array}\right.$$

Due to the proximity of the sending and receiving nodes, the paper has chosen to utilize the free space model (where *d* < *d*_0_). To align with previous work [29], the transmission radius is set at 50 m. To simplify the analysis, we have converted the total energy into the transmission of 2 M bits of data. Additionally, in each round, a node will be required to send 1 k bits of original data every 5 min.

The minimum energy threshold for nodes to enter protected mode has been set at 10%. While in this mode, nodes are only permitted to send original data and are not capable of forwarding any packets. To determine the average lifetime of networks, which is the rate at which nodes die off by 50%, we analyzed the results of 100 simulations using 100 different random seeds for network deployment. We varied the number of nodes from 100 to 1000 to obtain distinct values. The comparison of the lifetime achieved by the four algorithms is displayed in Figure 6.

### 6.2. Performance Comparison

Figure 6 demonstrates that, as the number of nodes increases, the network’s lifetime is significantly extended. The DMHS algorithm necessitates more execution rounds when compared to the other two algorithms, and this distinction becomes more evident as the network grows. With over 500 nodes, the opportunity flooding algorithm fails to effectively extend the network’s lifetime due to a broadcast storm. Additionally, it is clear that the DMHS algorithm enhances the lifetime by more than 12% to 57% with over 700 nodes. As the number of nodes increases, the network lifetime of the other algorithms does not improve significantly beyond 500. This is because with the increase in the number of nodes, the network becomes more complex and frequent packets need to be sent between individual nodes leading to energy consumption. Once some of the nodes in the network fail, it results in a large number of packets being forwarded centrally from certain preferred nodes, which can progressively accelerate the death of the network. The DMHS algorithm selects the next hop node by probabilistic forwarding. As the number of nodes in the network increases, it will inevitably lead to an increase in the density of nodes, then the choice of paths becomes a lot, which can make the residual energy of each node in the whole network balanced.

In order to accurately replicate a real-world explosive event, our team utilized a network of 1000 nodes. By inducing the death rate of specific nodes through targeted invalidation and manual creation of black holes, we were able to achieve a more precise evaluation. Our primary metric for comparison purposes was the average delivery rate, which included the percentage of nodes unable to route to the sink despite being active. Our findings are presented in Figure 7 and demonstrate a clear correlation between the number of invalidated nodes and the average delivery rate.

The DMHS algorithm demonstrates improved average delivery rate performance as the network expands, with this advantage becoming more prominent when the death rate is between 50% and 70%. This is attributed to the algorithm’s ability to recursively explore other nodes by adjusting its hop count when a route is lost, as long as there is only one connected route to the sink. In contrast to traditional algorithms, the DMHS algorithm overcomes limitations such as becoming stuck in local optimal solutions and slower convergence. Consequently, it can be inferred that the DMHS algorithm exhibits reduced energy consumption and a high level of robustness.

We statistically analyzed the above results, first we used the ANOVA method to compare these four algorithms. The *p*-value for the ANOVA test is about 0.6817. The test results show that there is no significant difference in the mean values between the groups being compared. After analyzing the results, we believe that the reason for this phenomenon is because when the network structure is complete and the number of dead nodes is small, the various algorithms are able to achieve a better average delivery rate, which cannot reflect the advantages of DMHS. In order to better compare the differences in performance among algorithms, we use the sum of weighted average delivery rates metric to measure the strengths and weaknesses of the algorithms. The main idea is to sum the average delivery rates under all mortality rates multiplied by the mortality rate. The purpose of this is to lower the algorithm’s score for average delivery rates at low mortality rates, and the better it performs at high mortality rates, the higher the score. Figure 8 below shows the scores of the individual algorithms, and it can be seen that the DMHS algorithm has a 10% to 21% improvement compared to the other algorithms.

## 7. Conclusions

This paper presents a robust routing approach that leverages dynamic programming to address the challenges of unreliable links in WSNs. Designed for deployment in extreme environments that are prone to explosions, the proposed algorithm employs dynamic minimum hop selection to enable efficient routing without compromising battery life. By avoiding the recalculation of hop status through iterative and recursive calls, the dynamic programming method has low time complexity. Additionally, the algorithm’s dynamic adjustment of hop stats ensures energy balance, overcomes the problem of local minimum in traditional algorithms, and addresses the black hole problem.

Simulation results reveal that the proposed Dynamic Minimum Hop Selection (DMHS) algorithm has a low routing cost and significantly improves delivery performance in environments with unreliable links. The effectiveness and feasibility of the proposed approach in dealing with routing issues in extreme environments make it particularly well-suited for deployment in volatile settings. By employing probabilistic forwarding decision-making and distributed route discovery, the algorithm addresses the unique challenges posed by high-risk and unstable environments, enhancing the network’s lifespan and delivery rate.

The potential for WSNs to operate efficiently and robustly in demanding contexts is an essential requirement for critical applications in chemical plants, coal mines, nuclear power plants, battlefields, and other hazardous settings. The DMHS algorithm serves as a pivotal tool for such applications, providing a viable solution for ensuring efficient and reliable communication in severely challenging environments. We notice that the dynamic programming approach may encounter higher latency and complexity issues when the node density increases and the number of neighboring nodes is large, especially during initialization. Future research can investigate and address scalability challenges by utilization of clustering strategies and sleep scheduling in large-scale networks to effectively reduce node density.

## Figures and Tables

**Figure 1 sensors-23-07223-f001:**
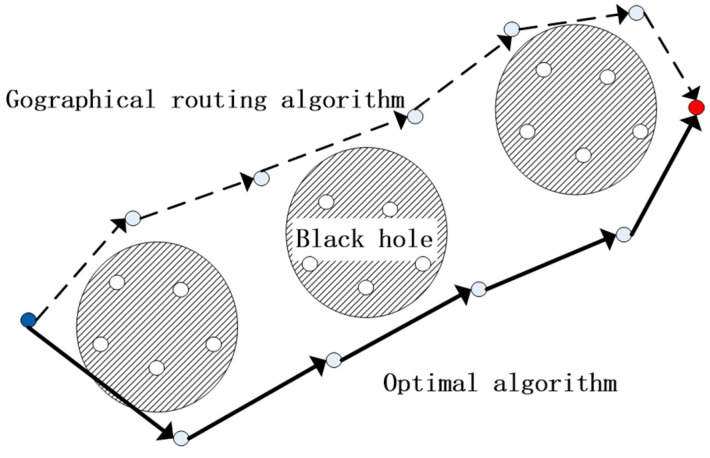
The local minimum value problem based on geographical location information.

**Figure 2 sensors-23-07223-f002:**
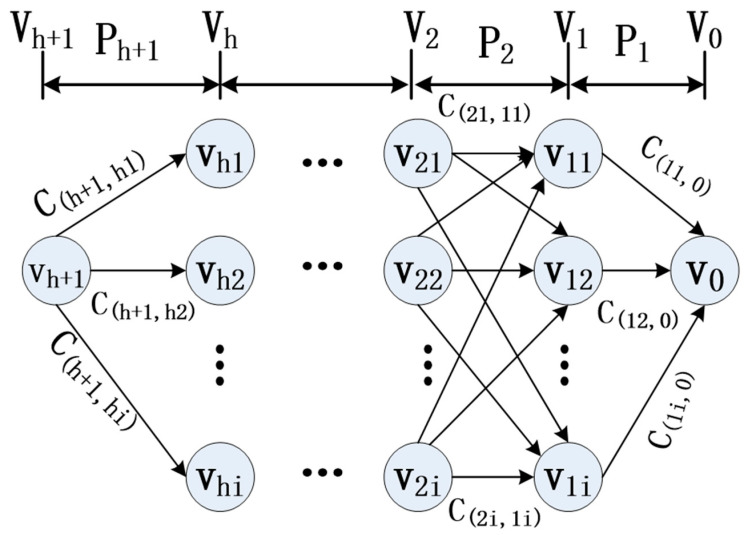
The hierarchical dynamic programming model.

**Figure 3 sensors-23-07223-f003:**
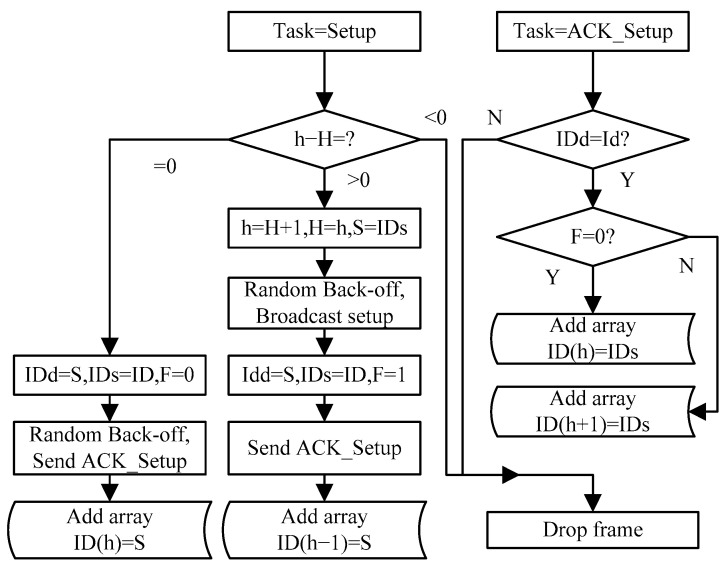
The flowchart for establishing a routing table.

**Figure 4 sensors-23-07223-f004:**
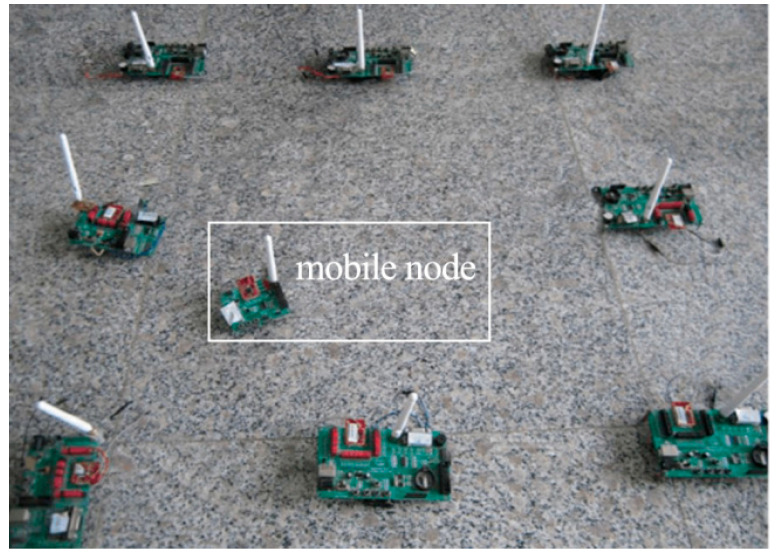
Experiment setting.

**Figure 5 sensors-23-07223-f005:**
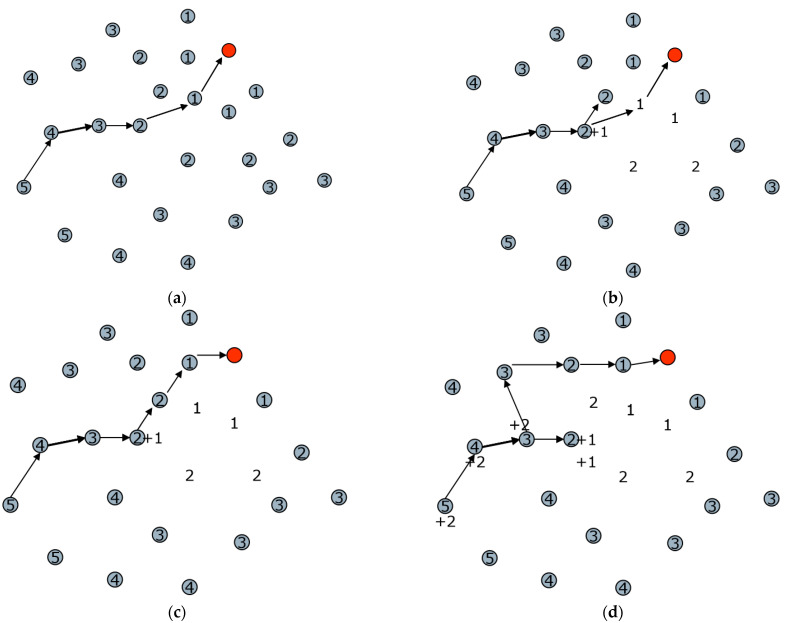
Examples of dynamic adjustments. (**a**) When the initial routing table is created. (**b**) When the penultimate hop node does not work. (**c**) A new route is established. (**d**) When another node on the new path is also out of work. (**e**) After iteratively adjusting the value of the relevant variable h, a unique path connecting to the sink can be found as long as it exists. (**f**) Dynamic adjustment of the optimal path. (**g**) Newly joined mobile nodes broadcast join frames. (**h**) A new path is finally established.

**Figure 6 sensors-23-07223-f006:**
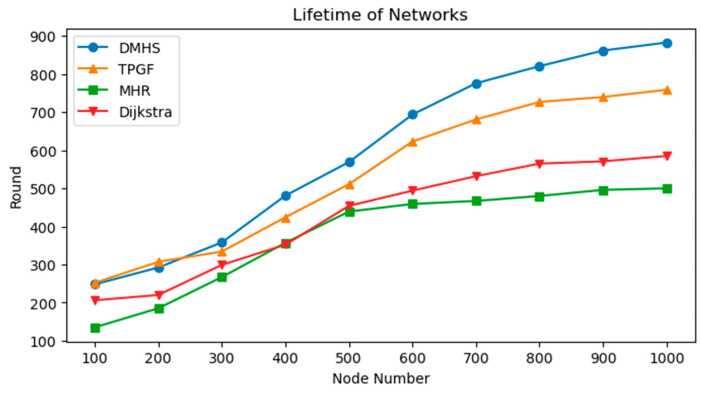
Comparison of the lifetime.

**Figure 7 sensors-23-07223-f007:**
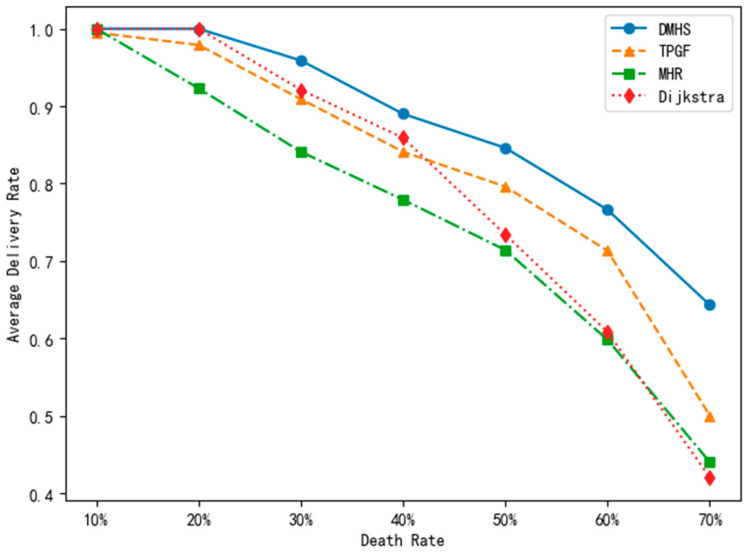
The comparison of average delivery rates.

**Figure 8 sensors-23-07223-f008:**
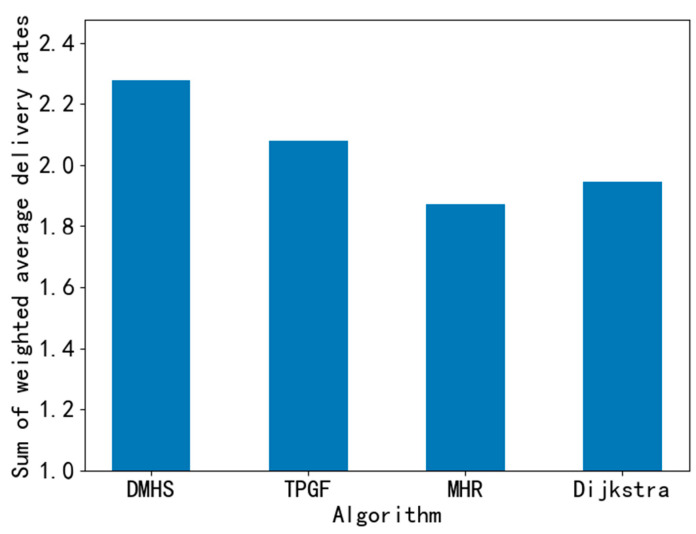
Sum of weighted average delivery rates.

**Table 1 sensors-23-07223-t001:** Comparison of previously used algorithms.

Routing Algorithms	Delivery Ratio	Energy Efficiency	Geographic Information	Robustness	Time Complexity
DMHS	high	high	unnecessary	high	O(n × m)
TPGF	middle	high	required	middle	O(n)
MHR	low	low	unnecessary	low	O(n + E)
Dijkstra	middle	middle	unnecessary	high	O(n^2^)

**Table 2 sensors-23-07223-t002:** The format of data packet.

Routing Table Entries		
Head	ID	ID number of this node
	ID_S_	ID number of previous node
	ID_D_	ID number of next node
	H	Current number of hops from the node to the sink
	Flag	Used to trace back the previous node
Body	Task-type	Normal, emergency, join, and quite task type
	Src	Source node address
	Dest	Destination node address
	Data	Data to be transmitted
	Checksum	Used as error control

## Data Availability

The data presented in this study are available on request from the corresponding author. The data are not publicly available due to their sensitive nature originating from Maoming Petrochemical Plant, which prohibits their sharing for confidentiality reasons. The data contain proprietary information owned by the plant and their distribution may infringe upon trade secrets and compromise the plant’s competitive advantage in the industry. Access to the data is strictly controlled by the plant administration, and any external dissemination poses a significant risk to their operations and security.

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
