# Peer review of "Innovative DMHS Algorithm Application in Wireless Sensor Networks for Efficient Routing in High-Risk Environments"

_sensors, 2023, doi:10.3390/s23167223_

Round 1

Reviewer 1 Report

The manuscript presented by the author's Ma and Deng deals with a robust routing approach that leverages dynamic programming to address the challenges of unreliable links in wireless sensor networks. The main issue of the manuscript is that very little was done to apply the new algorithm. Although the author reports present solid research, they should explain the following remarks in detail.

- The introduction is divided into two parts: 1. Introduction and 2. Related work, therefore the introduction is very detailed and quite extensive compared to the rest of the manuscript.

- In the introduction – the authors should suggest the application of the described algorithm, link with the new possibilities, and support with references.

- Summarize your innovations for the algorithm: a table comparing the main characteristics of the newly developed system containing the most significant data about the previously used systems should be introduced in the manuscript.

- Line 400: “This section is not mandatory but can be added to the manuscript if the discussion is unusually long or complex.“ Comment: ?

- Did any tests using real three-dimensional or hazardous sensor network signals confirm the usefulness of the presented system?

- The abbreviation Wireless Sensor Networks (WSNs) is described in several places in the text, but not at the first mention.

- In Line 153. The authors used all capital letters in the name of the author.

- What are the benefits of this approach in malicious attacks?

- I can't help but notice that in some parts the manuscript is arranged similarly to the published paper: An Efficient and Reliable Algorithm for Wireless Sensor Network. I ask the authors to clarify.

- Several typos

Minor editing of the English language is required.

Author Response

Response to Reviewer 1 Comments

Dear reviewers

Re: Manuscript ID: sensors-2549572 and Title: Innovative DMHS Algorithm Application in Wireless Sensor Networks for Efficient Routing in Volatile Environments

Thank you for your letter and the reviewers’ comments concerning our manuscript. Those comments are valuable and very helpful. We have read through comments carefully and have made corrections. Based on the instructions provided in your letter, we uploaded the file of the revised manuscript.

We would love to thank you for allowing us to resubmit a revised copy of the manuscript and we highly appreciate your time and consideration.

Sincerely.

Yuanjia.Ma

Point 1: The introduction is divided into two parts: 1. Introduction and 2. Related work, therefore the introduction is very detailed and quite extensive compared to the rest of the manuscript.

Response 1: Thank you for your feedback on the structure of our manuscript's introduction. We acknowledge that the introduction is divided into two sections, namely "Introduction" and "Related work". This division has been deliberate to provide a comprehensive understanding of the research context and establish the novelty of our work. We acknowledge that the introduction section may be longer in comparison to other parts of the manuscript. Consequently, we have undertaken moderate streamlining to eliminate superfluous content while expanding on other section of the thesis, resulting in a more rational structure.

Point 2: In the introduction – the authors should suggest the application of the described algorithm, link with the new possibilities, and support with references.

Response 2: In the introduction of paragraph 5&6, we have highlighted the potential application of the described algorithm in various real-world scenarios. Furthermore, we have supplemented our claims with references to related studies and research that have successfully employed similar algorithms in practical applications. This approach described algorithm can further expand the current possibilities within the field.

Point 3: Summarize your innovations for the algorithm: a table comparing the main characteristics of the newly developed system containing the most significant data about the previously used systems should be introduced in the manuscript.

Response 3: In response to your suggestion, we have made revisions to the manuscript in the final paragraph of the introduction chapter to incorporate a table that provides an overview of the essential characteristics and significant data pertaining to the previously utilized systems. This table will enhance the clarity and conciseness of the manuscript by providing a comprehensive overview of the advancements made in our algorithm.

Point 4: - Line 400: “This section is not mandatory but can be added to the manuscript if the discussion is unusually long or complex.“ Comment: ?

Response 4: We express our sincere gratitude to the reviewers for their valuable corrections. We deeply apologize for the inadvertent inclusion of annotate with comments in the text during our latest revision. This was an oversight on our part, we have diligently removed the annotations that should not have been present in the body of the text.

Point 5: Did any tests using real three-dimensional or hazardous sensor network signals confirm the usefulness of the presented system?

Response5: Indeed, we conducted a practical field deployment during the initial phase to validate the algorithm's feasibility. Figure 5 depicts the experiment performed through actual field deployment, presented in schematic form. Nonetheless, due to site restrictions and limited sensor availability, we solely concentrated on verifying the algorithm's feasibility, while the performance evaluation was carried out through simulation. In the revised draft, we have included pertinent real test part in section 5.

Point 6: The abbreviation Wireless Sensor Networks (WSNs) is described in several places in the text, but not at the first mention.

Response6: I would like to express my sincere gratitude to the reviewer for providing valuable corrections. I have incorporated these changes into the manuscript, retaining the full title as it was initially mentioned.

Point 7: In Line 153. The authors used all capital letters in the name of the author.

Response7: Upon the request of another reviewer, the introduction section requires streamlining and revision, leading to the removal of the previous references 18 and 19.

Point 8: What are the benefits of this approach in malicious attacks?

Response8: Many thanks to the reviewers for their questions. We appreciate the opportunity to further elaborate on the advantages of our method in malicious attacks. Our method has several advantages in malicious attacks:

  1. Strong Robustness: Our approach addresses scenarios of node failure in high-risk environments by employing a Dynamic Minimum Hop Selection routing algorithm. Regardless of the cause of node failure (be it blast damage, energy depletion, or malicious attacks), our system efficiently identifies such failures as long as at least one connectivity link exists in the environment. Consequently, it exhibits strong robustness against various forms of malicious attacks.
  2. Timely Detection and Response: Although our approach cannot ascertain the precise cause of node failure, the damage inflicted by malicious attacks enables the algorithm to automatically bypass the failed node and promptly adjust the path. In comparison to other algorithms relying on fixed location information, hop count, and distance measurements, our method demonstrates reduced susceptibility to damage.

Point 9: I can't help but notice that in some parts the manuscript is arranged similarly to the published paper: An Efficient and Reliable Algorithm for Wireless Sensor Network. I ask the authors to clarify.

Response9: Regarding the published paper, I would like to clarify the situation. The papers you mentioned is indeed a paper I presented in a previous international conference, which is the preliminary results of our research, and these results have been successfully applied for patents. Now, we have further extended and enriched our research on this basis, tracked the latest research progress, done actual field deployment, and further demonstrated the feasibility of the algorithm.

Point 10:  Several typos.

Response10: I have scrutinized and corrected the paper to ensure that all typos have been corrected in a timely manner. With your guidance, I will further enhance my grammar and spell checking of the paper to ensure that the final version is free from any errors.

Reviewer 2 Report

This paper presents a routing algorithm for wireless sensor networks (WSNs) called Dynamic Minimum Hop Selection (DMHS), designed for operation in extreme, volatile environments. The DMHS algorithm leverages dynamic programming to optimize routing and manage unreliable links, overcoming common issues such as local minimum traps and black holes prevalent in traditional algorithms. Simulations indicate that the DMHS algorithm enhances the delivery performance and extends the network's lifespan, offering improved energy efficiency. This robust, adaptive algorithm holds potential for deployment in high-risk environments, like chemical plants, coal mines, nuclear power plants, and battlefields. Future research aims to explore the application of this method in more complex network scenarios and address the issue of black holes in disaster regions.

The authors articulate a complex problem clearly, propose a novel solution, and provide a comprehensive evaluation of their Dynamic Minimum Hop Selection (DMHS) algorithm. The use of a robust simulation environment and comparisons with other existing solutions add credibility to the claims made about the DMHS algorithm. Their work is well-structured, and the literature review offers a solid foundation for the study. However, the paper could be improved by further considering edge cases or potential limitations of the proposed algorithm, as well as elaborating on possible strategies to remedy them. Future research directions are stated, but more detailed explanations would be beneficial. Overall, the paper represents a good contribution to the field of wireless sensor network routing protocols.

Major comments:

1. The paper predominantly bases its simulation scenarios on ideal or predictable conditions, such as assuming a uniform energy value for each node. However, real-world energy consumption in a network could significantly vary due to factors such as distance to the receiver, data rate, and interference.

2. The performance evaluation of the algorithm is solely reliant on simulations. While simulations provide valuable initial testing, actual field deployment may exhibit varied results due to unforeseen factors or assumptions taken during simulation.

3. The study lacks a thorough exploration of edge cases and potential limitations of the proposed algorithm. For example, it fails to fully articulate the implications of multiple simultaneous failures, the existence of mobile nodes, or the routing impact of physical barriers.

4. The comparative analysis of the proposed algorithm is restricted to only two other algorithms. A broader comparison, including algorithms focusing on similar problem domains, would offer a more comprehensive evaluation.

5. The paper provides insufficient discourse about the practical implementation of the algorithm, including facets like computational complexity, memory requirements for the routing tables, and the overhead the algorithm might introduce.

6. The energy consumption model employed is inadequately described and justified. A comprehensive explanation would lend greater credibility to the simulation outcomes, and discussing the trade-off between energy consumption and network lifetime would enrich the analysis.

7. The assumption that network nodes are predominantly static and possess uniform transmission radius does not reflect realistic scenarios, such as mobile nodes or nodes with varied capabilities, frequently found in wireless sensor networks.

8. The paper would benefit from a more stringent statistical analysis of the simulation results. In its absence, it becomes challenging to ascertain the significance of the reported improvements.

9. The scalability aspect of the proposed algorithm is not satisfactorily addressed. With an increasing network size, the dynamic programming approach may encounter higher latency and complexity issues.

10. The algorithm's effectiveness largely depends on accurate hop counting. Any corruption or miscalculation of hop count could adversely impact the algorithm's performance, a point overlooked in the study.

Author Response

Response to Reviewer 2 Comments

Dear reviewers

Re: Manuscript ID: sensors-2549572 and Title: Innovative DMHS Algorithm Application in Wireless Sensor Networks for Efficient Routing in Volatile Environments

Thank you for your letter and the reviewers’ comments concerning our manuscript. Those comments are valuable and very helpful. We have read through comments carefully and have made corrections. Based on the instructions provided in your letter, we uploaded the file of the revised manuscript.

We would love to thank you for allowing us to resubmit a revised copy of the manuscript and we highly appreciate your time and consideration.

Sincerely.

Yuanjia.Ma

Point 1: The paper predominantly bases its simulation scenarios on ideal or predictable conditions, such as assuming a uniform energy value for each node. However, real-world energy consumption in a network could significantly vary due to factors such as distance to the receiver, data rate, and interference.

Response 1: We acknowledge the use of idealized assumptions in our study to simplify the simulation model. These assumptions facilitate quantitative evaluation of algorithm performance in a controlled environment. However, we recognize that these factors can significantly impact energy consumption in real-world applications. In fact, the previous section 5 was the experiment we did based on actual field deployment and then presented in schematic form. The nodes arranged in accordance with real-world scenarios, and additional factors including node distances, data rates, and interference will be taken into account to enhance the accuracy of the real-world environment simulation. By doing so, we aim to ensure the practical feasibility and reliability of our findings.

Point 2: The performance evaluation of the algorithm is solely reliant on simulations. While simulations provide valuable initial testing, actual field deployment may exhibit varied results due to unforeseen factors or assumptions taken during simulation.

Response 2: We fully understand and acknowledge this viewpoint. In fact, we conducted actual field deployment in the early stage to validate the feasibility of the algorithm. At the same time, we have highlighted the reasons and limitations of using simulation to evaluate algorithm performance. We emphasize that actual deployment may be influenced by various unpredictable factors, which cannot be fully simulated in the simulation. In our research, to overcome limitations in terms of site and nodes count, we employed simulation as an effective approach to assess algorithm performance. Through simulation, we were able to test the algorithm in a controlled environment and obtain quantitative preliminary performance indicators.

Point 3: The study lacks a thorough exploration of edge cases and potential limitations of the proposed algorithm. For example, it fails to fully articulate the implications of multiple simultaneous failures, the existence of mobile nodes, or the routing impact of physical barriers.

Response 3: Regarding your mention of the shortcomings of our paper in exploring the boundary cases and potential limitations of the proposed algorithm, we would like to respond to this. In the revised version, we have fully considered the boundary cases and potential limitations you have raised and explored them in detail. The goal of our proposed algorithm is to address the shortcoming that nodes are prone to failure in high-risk environments. Multiple simultaneous failures in such situations are certainly frequent. In the experiment in Fig. 6, we further investigate the impact on our algorithm when 3 failures occur simultaneously. It should be explained that we have not considered the case where all beacon nodes are mobile for the time being. Although our algorithm does not need to provide information about the locations of the nodes, the locations of the nodes need to be fixed from the beginning. The mobile nodes described in the paper are those carried by robots or UAVs involved in network repair during rescue mission.

Point 4: The comparative analysis of the proposed algorithm is restricted to only two other algorithms. A broader comparison, including algorithms focusing on similar problem domains, would offer a more comprehensive evaluation.

Response 4: We agree your suggestion for a broader comparison encompassing algorithms focusing on similar problem domains. We have expanded our comparative analysis accordingly in order to provide a more comprehensive evaluation of the proposed algorithm.

Point 5: The paper provides insufficient discourse about the practical implementation of the algorithm, including facets like computational complexity, memory requirements for the routing tables, and the overhead the algorithm might introduce.

Response 5: We apologize for overlooking these crucial details and appreciate your suggestion to include them in our paper. In our revised version, we will provide a comprehensive analysis of the algorithm's computational complexity, outlining the time complexity involved in its implementation. We will also address the memory requirements for the routing tables and discuss any potential overhead that might arise from utilizing the algorithm. Chapters 4.4 and 4.5 in revised version show the details.

Point 6: The energy consumption model employed is inadequately described and justified. A comprehensive explanation would lend greater credibility to the simulation outcomes, and discussing the trade-off between energy consumption and network lifetime would enrich the analysis.

Response 6: Thank you for your valuable feedback. We acknowledge the concern regarding the inadequate description and justification of the energy consumption model employed. A comprehensive explanation of the model has been added in the manuscript section 3.2 to enhance the credibility of the simulation outcomes.

Point 7: The assumption that network nodes are predominantly static and possess uniform transmission radius does not reflect realistic scenarios, such as mobile nodes or nodes with varied capabilities, frequently found in wireless sensor networks.

Response 7: Indeed, the assumption that network nodes are predominantly static and possess uniform transmission radius, as presented in our study, may not reflect certain scenarios such as those involving mobile nodes or nodes with varied capabilities as you have pointed out.

However, the assumption of predominantly static nodes with uniform transmission radius was made intentionally in our study as it corresponds to the high-risk scenario we aimed to investigate. We recognize that the scenario involving majority of the nodes being mobile might require a different algorithm, possibly adopting the opportunistic networks, but it typically exhibits high latency, which may not be suitable for high-risk monitoring and alarm scenarios that require real-time or near-real-time responses.

Our focus in this work is on the scenarios where nodes are predominantly static, with a few mobile nodes serving as repair functions, as it offers a more stable and predictable network structure conducive to real-time monitoring and control. We anticipate that our work can form the foundation for further investigations into more complex scenarios that involve mobile nodes or nodes with varied capabilities.

We will clarify this point in the revised manuscript to avoid any confusion regarding the assumptions made in our study.  

Point 8: The paper would benefit from a more stringent statistical analysis of the simulation results. In its absence, it becomes challenging to ascertain the significance of the reported improvements.

Response 8: To address this concern, we performed additional statistical tests on the simulation data in section 6.2 to assess the significance of the reported improvements. We take the Sum of weighted average delivery rates methods, to analyze the data and provide a more comprehensive evaluation. This will help strengthen the reliability and validity of our results.

Point 9: The scalability aspect of the proposed algorithm is not satisfactorily addressed. With an increasing network size, the dynamic programming approach may encounter higher latency and complexity issues.

Response 9: Thank you for providing valuable feedback on the scalability aspect of our proposed algorithm. We acknowledge the importance of scalability in wireless sensor networks and understand your concerns regarding the applicability of the dynamic programming approach in larger networks. We agree that as the node density increases and the number of neighboring nodes grows, especially during initialization, the dynamic programming approach may encounter higher latency and complexity issues.

However, we believe that our algorithm still holds practical relevance in scenarios involving small and medium-sized networks. Additionally, it is important to note that our study represents an initial exploration of the effectiveness of the proposed algorithm. We genuinely appreciate your valuable opinions and suggestions, which have significantly enhanced the clarity and comprehensiveness of our work. Moreover, they have inspired new ideas for our future research.

In future research, we aim to address scalability challenges by exploring the utilization of clustering strategies and sleep scheduling in large-scale networks to effectively reduce node density.

Point 10: The algorithm's effectiveness largely depends on accurate hop counting. Any corruption or miscalculation of hop count could adversely impact the algorithm's performance, a point overlooked in the study.

Response 10: You have rightfully pointed out that any corruption or miscalculation of hop count could have adverse effects on the algorithm's performance, and we acknowledge the importance of accurate hop counting in wireless sensor networks, as it plays a crucial role in determining the distance or proximity between nodes.  

In our revised manuscript, we included some dedicated part in section 3.4 that discusses the potential sources of errors in hop count estimation and their implications on the algorithm's performance. We also explored possible mitigation strategies, such as incorporating error correction techniques or retransmission to enhance the accuracy of hop count calculations.

Round 2

Reviewer 1 Report

 The manuscript can be accepted in its present form.

The presented English level is acceptable.

Reviewer 2 Report

Thanks for addressing my comments.